# Why So Serious about Foreign Capital?

**Ashish Kumar Sedai**

Department of Economics, Colorado State University, Fort Collins, CO 80523, USA; ashish7@colostate.edu;
Tel.: +1-970-690-3511

**Abstract:** This study examines the cost and benefits of capital inflow in emerging economies and delineates equity and debt to examine the nature and trends of capital inflows in Brazil, Russia, India, China, South Africa (BRICS), East Asia and Sub-Saharan Africa since their economic reforms. We adopt a two-step process to address endogeneity and to tease out the causal effect of capital flow on economic growth and vice versa. First, we run the panel Granger causality test to examine the precedence of causality between per capita GDP growth, Foreign Direct Investment (FDI) inflows, portfolio inflows and the real effective exchange rate. We follow this test with a fixed-effect panel regression model to test for the magnitude of causality between the variables. The study finds the presence of a strong causality between FDI equity flows and a weak and lagged causality between short term capital flows and economic growth. In the short-run, there is bi-directional causality in growth and equity flows. In the longer run, the effects of equity fade away, but the effect of sustained debt kicks in. Among other results, an average currency appreciation for one-year causes equity inflow and causes GDP growth for two years.

**Keywords:** FDI; portfolio; economic reforms; panel Granger causality; BRICS; Granger causality test

## 1. Introduction

Post-World War II, the idea of a big push (Roland 1943) in maintaining a positive investment climate has rocketed as a key policy tool for economic growth, development and poverty reduction. In the past few decades, the global financial system has witnessed unprecedented level of capital flows, as well as cross-border economic, financial and business integration both within advanced and emerging economies (World Bank 2016). Some economists have claimed that in a rapidly growing global financial system, improving the opportunities and incentivizing firms to invest, create jobs, and to expand should be a top priority for governments in emerging economies (Iamsiraroj and Doucouliagos 2015). Others have argued that the key to sustainable growth is not just about increasing the quantum of investments, they are also about spurring productivity improvements (Dollar 2005; Brainard et al. 2016), and therein the role of stable FDI and equity flows versus short term debt flows kick in. Empirical evidence has shown that emerging economies have had winners, losers, and victims of financial crises following their reforms and integration with the global financial system (Blecker 1999). Thus, a wide-ranging debate has opened up on how to design a "new architecture" of investment regimes to garnish the benefits of capital flows.

The discussion above points to an important debate, does economic growth lead to capital inflow or is it the other way around? The understanding of this could lead to either a top down or a bottom up approach in constructing policy tools for capital flows in emerging economies. In the recent past, capital flows have had both positive and negative effects, there have been scores of studies on crisis in East Asian and Latin American economies which have attributed unregulated and unstructured capital flows to be a major concern. Economies have gone on to the extent of ignoring the need for fiscal stimulus to keep the door for capital to flow in. The agenda set by the Washington Consensus

and pushed by major international institutions have created a hangover of capital flow in emerging economies, and it is important for these economies to set the agenda straight on what is the ultimate objective of attracting and acquiring capital at the policy level. Thus, it would be interesting to see if there is bi-directional '*causality*' between economic growth and capital inflow in emerging economies. It also has enduring policy relevance as developing countries try to decide whether to open themselves up more to financial globalization, and if so, in what form and to what degree. This questions in retrospect and the future provides a guideline for policy makers in determining the level of exchange rate adjustments, capital controls and prudential regulations required for macro-economic stability. does foreign capital play a helpful, benign, or malign role in economic growth?

These empirics point to a pertinent question: What is the effect of capital flow on economic growth? Does foreign capital drive economic growth? Is economic growth driving capital inflow? Are there definitive causal impacts of these variables on each other? There has been a passionate debate among economists, policymakers, and members of civil society about the first few questions. However, thus far, there has been no study on the causal impact of foreign capital on economic growth and vice versa. This question has gained importance in recent years because of the curious, even seemingly perverse, phenomenon of global capital flowing "uphill" from poorer to richer countries (Prasad et al. 2007). However, it has economic relevance beyond the current conjuncture because it goes to the heart of the process of development and the role of foreign capital in it. It also has enduring policy relevance as developing countries try to decide whether to open themselves up more to financial globalization, and if so, in what form and to what degree. These questions, in retrospect and the future, provide a guideline for policy makers in determining the level of exchange rate adjustments, capital controls and prudential regulations required for macro-economic stability. Does foreign capital play a helpful, benign, or malign role in economic growth?

Part II of this paper would analyse the background, costs and benefits of financial liberalization with specific focus on capital inflows in India and China as the leaders of developing economies. We discuss the literature in terms of the investment regimes and economic reforms in developing countries, especially after the financial crisis of 2008 and point out the commonalities in the financial sickness of developing countries. We then evaluate the relative effectiveness and efficiency of their regimes. Part III presents the model and the variables used for the study. Part IV examines the summary statistics, the results of the Dumitrescu and Hurlin (2012) Granger causality test and the panel data fixed effect regressions. Part V concludes with discuss potential avenues for reforms

## 2. Literature Review

Two decades ago, liberalization became a prerequisite step to realize the growth potential of the emerging economies, which had long been clustered under the ambit of protectionist policies. Domestic saving constraints prompted the deregulation of inefficient growth retarding investment regimes, and opening up the economies to the global capital became the mantra. However, increasing openness and dissolving regulations came with politico–economic opportunity costs. These costs were both explicit and implicit. Explicitly, dissolving regulations affected the fostering of infant industries in host economies (Calvo 2007). Before 1991, there was a strong political support for infant industry protection in India, which restricted the flow of foreign capital in sectors currently categorized under an automatic route (Melitz 2005). Implicitly, increased openness makes an economy vulnerable to global economic shocks (Taylor 1998), capital flights (Blecker 1999), de-nationalization (Patnaik 2003), stunted economic growth (Calvo 2007) and rising inequality (Dollar 2005). There are commonalities of mismanagement of capital linked to both the East Asian economic and the Latin American crises—unwarranted levels of openness and the flow of short term debt flows (Calvo 2007), also called the hot money (Patnaik 2003).

Since liberalization and accession to World Trade Organization (WTO) in terms of opening up, there has been an extraordinary inflow of foreign capital which has been the driving force behind the economic growth of India and China (El-Erian 2008). It is remarkable to note that both India and China, despite being leaders in capital inflows, slipped through the global financial crisis least

harmed, maintaining an above-average economic growth. Sweeney (2010) argued that these growth miracles were a result of the measured and prudentially regulated yet profound liberalization of foreign investments, beginning with Chinese economic reform in 1978 and later followed by India in 1984 with the doing away of License Raj and the New Economic Policy 1991. However, unlike India and China, other emerging (peripheral) economies experienced drastic growth effect following their sensitivity to financial flows (Aizenman et al. 2016). The tequila crisis (1994) and the East Asian Tiger crisis (1997) are among the major crisis that were initiated by sudden stops in capital flows. Figure 1 illustrates the absolute quantum of the FDI inflows in India and China in 2015; India attracted more FDI than China in core industries. Figures 2 and 3 show some interesting scissor-like trends. In 2015, India took over China as the leading destination for FDI inflows (Intelligence 2016), and in the same year, the trajectory of their GDP growth crossed paths, with China's GDP growth falling as India's rising.

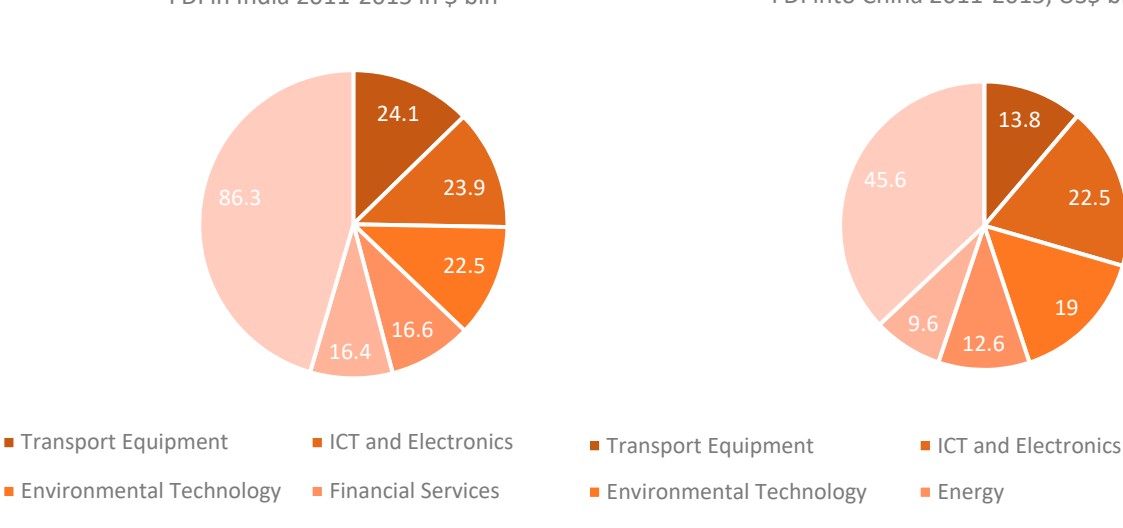

**Figure 1.** FDI Inflows in India and China 2011–2015, US$ billion. Source: Own elaboration using FDI markets data.

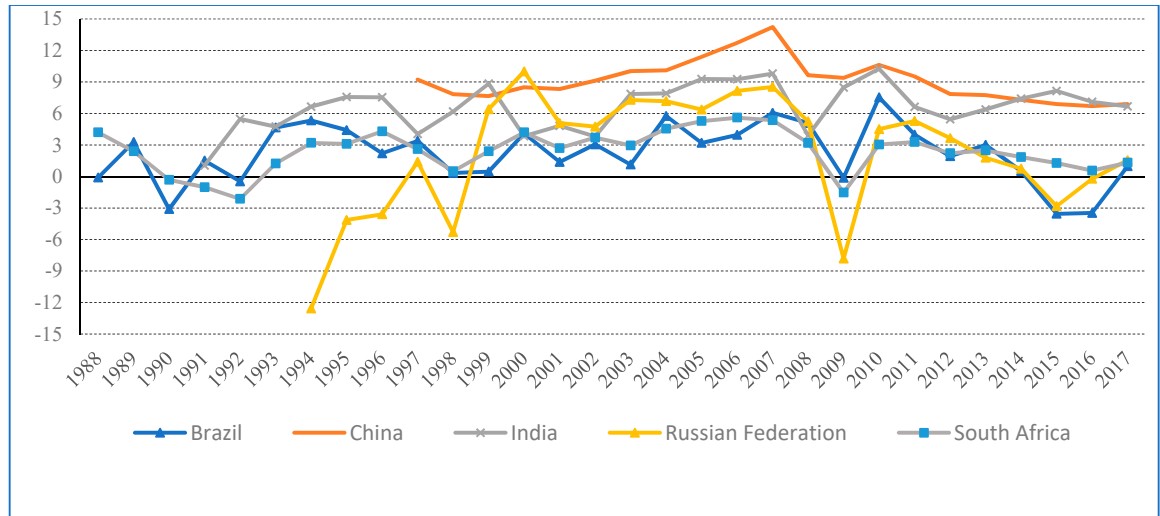

**Figure 2.** Real GDP (current US$), annual % growth. Source: Own elaboration based on World Development Indicators (World Bank).

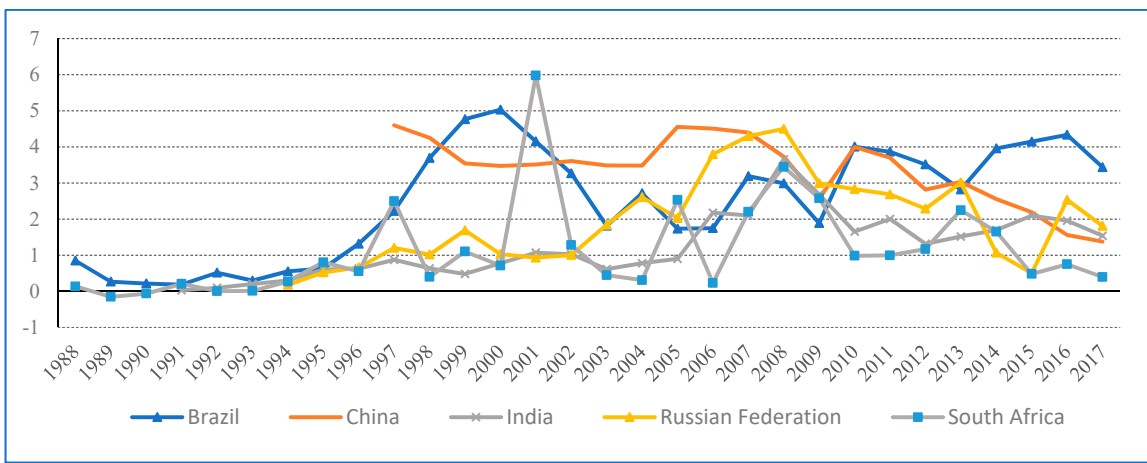

**Figure 3.** FDI, net inflows (as % of GDP). Source: Own elaboration based on World Development Indicators (World Bank).

The opening up of economies is a quest for modernity, awareness, information, poverty, the alleviation of life spans and living standard improvements, and the carrying out of political and social reforms (Frieden and Lake 2000). In this age of globalization, no country can live in isolation, and the ones which have tried have had abysmal performance in human development (Park 2004). The idea that market liberalization always produces benefits is erroneous in many respects (Schneider 2007). Policy regimes are contextualized under a broad socio–economic and political realm, and they are thus are inseparable. Contemporary economic research has noted the impact of certain characteristics and threshold requirements for smooth flow of finance in developing economies. These factors determine the desirable level of capital flow, where it should be employed, and how effective it will be (Iamsiraroj and Doucouliagos 2015). Financial flows are crucial in determining the level of development when gains from opening up can be harnessed.

The East Asian crisis and the recent recession of 2008 are illustrations of economies experiencing capital flight and the slowdown of economic growth without systematic investment regimes and unwarranted levels of influx of foreign dollars (Miller 2009). Attracting hot money (short term foreign capital) could temporarily fuel economic growth but is vulnerable to quick outflow should there be an opportunity to migrate elsewhere (Patnaik 2003). Hence, among other forms of investment, an FDI acts as a strong buffer against global economic swings and creates larger employment opportunities; hence, it is a more effective investment, especially in labor-abundant, developing countries (Scholte 2005).

Figures 3–5 present data on the FDI, net inflows, portfolio flow and investor confidence in BRICS. There has been an interesting scissor-like trend in terms of FDI flows in India and China since 2010. It is remarkable to note that in the years leading up to 2015, China's real GDP growth grew at an average of 6.4% per annum from 2010 to 2015, while India registered an average real GDP growth of 7% during the same period. There has been no consistent trend among the BRICS economies in attracting FDI and portfolio flows, thus highlighting the changing positions in terms of being favorable destinations of capital inflows.

In 2016, both China and India achieved a 7.1% GDP growth. However, the forecast of growth made by international organizations put India ahead of China, at least for half a decade in the future (IMF 2016; World Bank 2016). There has been a wide-ranging debate on whether China's pain is India's gain. (Bagchi 2017). The slump in the Chinese stock market was partially due to the withdrawal of foreign funds, but these funds would find their destination in the next best alternative—India. Bagchi (2017) provided some recent empirical examples of outward capital flow from China to India in search for profits.

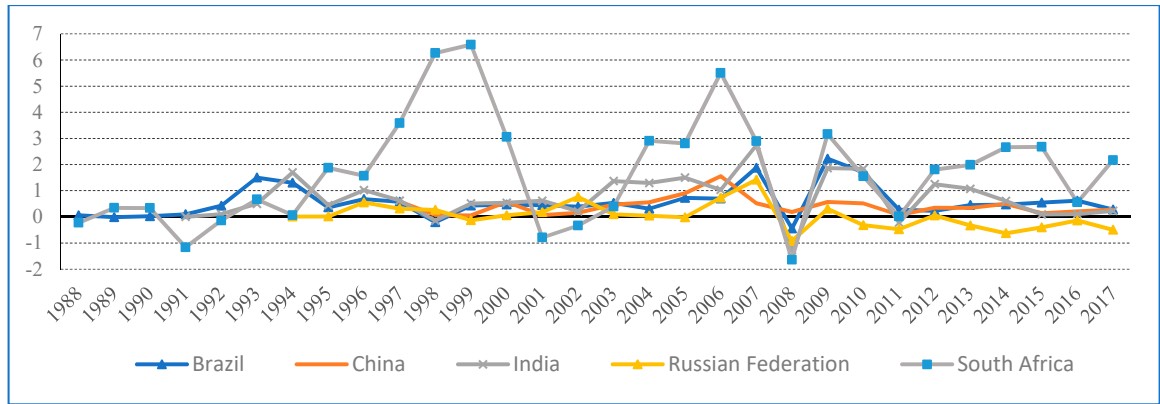

**Figure 4.** Portfolio, net inflows (as % of GDP). Source: Own elaboration based on World Development Indicators (World Bank).

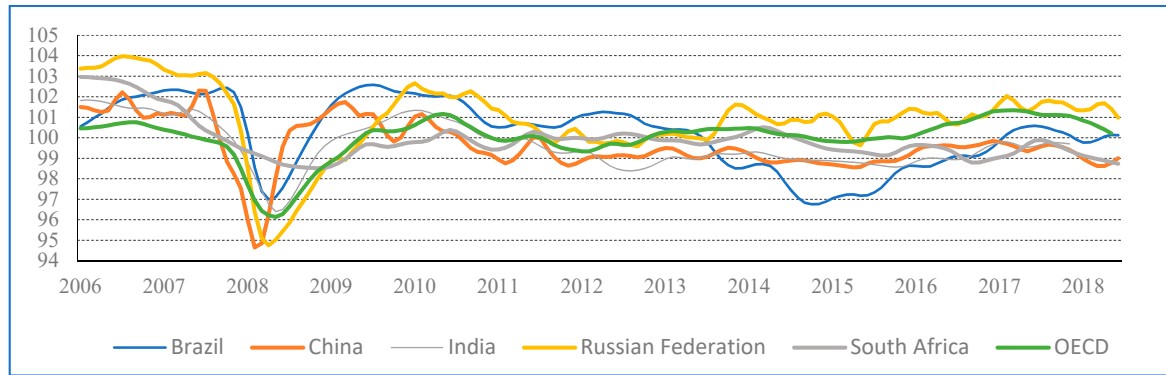

**Figure 5.** Investor confidence index. Source: Own elaboration based on World Development Indicators (World Bank).

FDI and equity flows are more stable forms of investment as compared to speculative investment flows in local securities markets and foreign institutional investments, especially in times of global economic panic. Hot money creates vulnerability, but a halt in FDI and equity flows can have painful implications for economic growth (Hanson 2009). A rapid reduction in these flows can have a devastating impact on communities and businesses that are largely dependent on them. However, the relationship between capital inflows and economic growth is far from being straightforward and varies across countries and time periods (Brainard et al. 2016). There have been scores of studies investigating the potential determinants of capital inflows, including the significance of sound investment climates and political stability (Schneider 2007), human capital in terms of educated workforce (Noorbakhsh et al. 2001), infrastructure (Wheeler and Mody 1992), trade openness, comparative labor costs, taxes and tariffs, and access to natural resources. In this context, India and China are yet to make the conditions for capital flows comparable with developed western and Organization for Economic Co-operation and Development (OECD) countries (World Bank 2016).

Policy regimes in each country emerge from a broad socio–economic and political context, and they are thus are inseparable. Contemporary economic research has noted the impact of certain characteristics and threshold requirements for smooth flows of finance in developing economies. These factors determine the desirable level of capital flow, where it should be employed, and how effective it will be (Iamsiraroj and Doucouliagos 2015). The upshot is that economic growth may not be the only way to improve the flow of capital, as there are many other factors that tangentially determine the flow of capital, such as the physical infrastructure of the economy, stable political climate, educated workforce and developed financial markets. The key element that differentiates developed economies from Emerging Markets (EMs) is in the very labeling of EMs, namely, the adjective emerging, especially if by emerging one means that these economies operate under highly incomplete information due

to, for example: (1) A lack of a sufficiently long track record and (2) weak economic and political institutions (Calvo 2007). These conditions make it more likely that, faced with a shock stemming from the international capital market, uninformed economic agents give more weight to the conjecture that shock has a large emerging market component, and they give less weight to the alternative, i.e., that shock comes from the international capital market.

The discussion above points to an important debate: Does economic growth lead to capital inflow, or is it the other way around? The understanding of this could lead to either a top-down or a bottom-up approach to constructing policy tools for capital flows in emerging economies. In the recent past, capital flows have had both positive and negative effects—there have been scores of studies on crisis in East Asian and Latin American economies which have attributed unregulated and unstructured capital flows to be a major concern. Economies have gone on to the extent of ignoring the need for fiscal stimulus to keep the door for capital to flow in. The agenda set by the Washington Consensus and pushed by major international institutions has created a hangover of capital flow in emerging economies, and it is important for these economies to set the agenda straight on what is the ultimate objective of attracting and acquiring capital at the policy level. Thus, it would be interesting to see if there is *'causality'* between economic growth and capital inflow in emerging economies.

## 3. Model

We used a panel Granger causality test to see if the movement of the covariates precedes each other in a times series. Though Granger causality predicts the precedence of a variable over another, it does not provide the magnitude of the effect. To tackle this issue, we ran a panel fixed-effects model to derive the magnitude of causality. The model followed in our analysis was introduced by Dumitrescu and Hurlin (2012). This test was a simple version of the Granger (1969) non-causality test for heterogeneous panel data models with fixed coefficients. It considers two dimensions of heterogeneity: The heterogeneity of the regression model used to test the Granger causality and the heterogeneity of the causality relationships.

$$y_{it} = \alpha_i + \sum_{k=1}^{K} \gamma_i^{(k)} y_{i,(t-k)} + \sum_{k=1}^{K} \beta_i^{(k)} x_{i,(t-k)} + \epsilon_{it}, \ i = 1, \ 2, \ 3, \ldots, N, t = 1, \ 2, \ 3, \ldots, T$$

which reduces to

$$y_{it} = \sum_{k=1}^{K} \beta_i^{(k)} x_{i,(t-k)} + U_{it}$$

Here, $U_{it} = \alpha_i + \epsilon_{it}$

In this study, the generalized (Dumitrescu and Hurlin 2012), model yielded the following

$$GDP_{it} = \alpha_i + \sum_{k=1}^{K} \gamma_i^{(k)} GDP_{i,(t-k)} + \sum_{k=1}^{K} \beta_i^{(k)} Foreign\ Capital_{i,(t-k)} + \epsilon_{it},$$

Alternately:

$$Foreign\ Capital_{it} = \alpha_i + \sum_{k=1}^{K} \gamma_i^{(k)} Foreign\ Capital_{i,(t-k)} + \sum_{k=1}^{K} \beta_i^{(k)} GDP_{i,(t-k)} + \epsilon_{it}$$

We captured the individual specific effects in the error term and considered $U_{it}$ is *iid* because we did not have the distribution of the error term. Here, $x$ and $y$ are two stationary variables observed for $N$ individuals in $T$ periods. $\beta_i = \left(\beta_i^1, \ldots, \beta_i^k\right)$, and the individual effects, $\alpha_i$ are fixed in the time dimension. Lag orders of $K$. are identical for all cross-section units of the panel. Autoregressive parameters $\gamma_i^{(k)}$ and the regression coefficients $\beta_i^{(k)}$ vary across groups.

We considered only the stationary outcome variable $y_{it}$. For each cross-section in panel $i = 1, 2, 3, \ldots, N$, at time $t = 1, 2, 3, \ldots, T$, $y_{it}$ is generated by the following AR (1) process.

$$y_{it} = \alpha_i + \rho_i y_{i,(t-1)} + \delta_i Z_{i,t} + \epsilon_{i,t}$$

Here, $Z_{i,t}$ is the exogenous variables in the model, $\alpha_i$ captures any individual specific effects, and $T$ is the time span of the panel. $N$ represents the number of cross-sections, $\rho_i$ is the autoregressive coefficients, and the error $\epsilon_{i,t}$ are assumed to be mutually-independent idiosyncratic disturbance. For $\rho_i < 1$, $y_{it}$ is stationary, and for $\rho_i = 1$, $y_{it}$ contains a unit root. We used the Pesaran and Smith, (1995) unit root test for panel data and used four lags to control for non-stationarity of the response variable.

$$y_{it} - y_{i,(t-1)} = \alpha_i + \beta_i(FDI_{i,t} - FDI_{i,(t-1)}) + \epsilon_{i,t}, \ i = 1,\ 2,\ 3,\ldots,N, t = 1,\ 2,\ 3,\ldots,T$$

Similarly:

$$FDI_{i,t} - FDI_{i,(t-1)} = \alpha_i + \beta_i\left(x_{it} - x_{i,(t-1)}\right) + \epsilon_{i,t}, \ i = 1,\ 2,\ 3,\ldots,N, t = 1,\ 2,\ 3,\ldots,T.$$

Under the null hypothesis, we assumed that there was no causal relationship for any of the units of the panel. This assumption is called the Homogeneous Non-Causality (HNC) hypothesis, which is defined as: $H_0 : \beta_i^{(k)} = 0 \ \forall i = 1,\ldots,N$. The alternative is specified as the Heterogeneous Non-Causality (HENC) hypothesis. Under this hypothesis, we allowed for two subgroups of cross-section units, and $\beta_i$. varied across groups. We used the Wald statistic for the null hypothesis $W_{N,T}^{HNC}$, which is related to homogeneous non causality condition with $N$ individual processes and $T$ as the time period.

$$W_{N,T}^{HNC} = \frac{1}{N}\sum_{i=1}^{N} W_{i,T}$$

Here, $W_{i,T}$ is the individual Wald statistics for the $i$th cross-section unit corresponding to the individual test of null hypothesis $H_0 : \beta_i = 0$. We decided on the existence of the Granger causality according to the value of the Wald statistic. We checked for robustness using a lag analysis of the Granger causality and panel regressions to check the magnitude of the effect of the covariate on the outcome variable $y_{it}$.

$$y_{it} = \alpha + \beta_i x_{i,t} + \rho_i Z_i{'} + U_{it}$$

with

$$U_{it} = \alpha_i + \epsilon_{it}$$

Here, $i$ is the country and $t$ is the year. $y_{it}$ and $x_{it}$ are capital inflow and GDP, respectively. $Z_i$ a set of control variables, and $\epsilon_{it}$ is the idiosyncratic error. All data on covariates were derived from the World Bank World Development Indicators (WDI) dataset. We collected yearly data from 1963 to 2018 for China, India, Brazil, Russia, South Africa, East Asia and Sub-Saharan Africa. The reason we included these developing economies was to have an observational cushion in doing the Granger causality and panel data analysis (see Appendix A Table A1). However, we did not have same number of observations for each country. Data were extracted on FDI inflow, portfolio investment inflow, GDP, real exchange rate, gross domestic capital formation, and gross domestic investment, and the data were more recent than old. The limitation of the data collection process was that a panel Granger causality requires a strongly balanced data set; thus, we lost a substantial amount of observations and degrees of freedom. We conducted a Hausman test to choose between random effects and fixed effects and have $p = 0.07$, which suggests the choice of fixed-effect over random effect at a 10% significance level. We computed the Z-bar and W-bar test statistics of the Homogenous Non Causality (HNC) hypothesis given by Dumitrescu and Hurlin (2012). Under the null hypothesis of HNC, there is no causal relationship for all the cross-units of the panel. Under the alternative, there is a causal

relationship from X to Y, at least for one cross-unit. The panel is balanced[1], and the W-bar statistic corresponds to the cross sectional average of the N standard individual Wald statistics of the Granger non causality tests. The Z-bar statistic corresponds to the standardized statistic for the fixed T sample.

## 4. Results

Tables 1 and 2 present the summary statistics on the key variables used for the study. Tables 3 and 4 provide the (Dumitrescu and Hurlin 2012) granger non causality test results to test the hypothesis of precedence. A preliminary glance at the summary statistics above shows that there have been differences across developing economics in terms of the share of the FDI in GDP and the share of net portfolios in GDP. The average of portfolio investments was negative, indicating a net outflow, although, as explained above, there has been an increase in net portfolio investment in recent years in India and China. The reason we created a structural break in the summary data was to understand how foreign and domestic capital evolved between World War II and recent economic reforms of these emerging economies.

**Table 1.** Summary statistics (Years: 1963–2018).

| | Sub-Saharan Africa | | | East Asia | | |
|---|---|---|---|---|---|---|
| | Years | Mean | Standard Deviation | Years | Mean | Standard Deviation |
| FDI (% of GDP) | 48 | 1.463 | 1.029 | 36 | 2.629 | 1.254 |
| Portfolio (% of GDP) | 0 | | | 0 | | |
| Nominal GDP growth rate | 48 | 3.379 | 2.589 | 36 | 8.002 | 1.979 |
| Gross Fixed Capital Formation (% of GDP) | 37 | 23.880 | 4.884 | 36 | 37.523 | 3.629 |
| Gross Domestic Savings (% of GDP) | 37 | 24.328 | 4.777 | 36 | 39.198 | 4.855 |
| | South Africa | | | China | | |
| FDI (% of GDP) | 48 | 0.813 | 1.201 | 36 | 2.843 | 1.676 |
| Portfolio (% of GDP) | 33 | −2.061 | 3.118 | 36 | −0.783 | 0.714 |
| Nominal GDP growth rate | 48 | 2.516 | 2.217 | 36 | 9.762 | 2.672 |
| Gross Fixed Capital Formation (% of GDP) | 48 | 22.165 | 5.096 | 36 | 40.302 | 4.509 |
| Gross Domestic Savings (% of GDP) | 48 | 23.365 | 5.437 | 36 | 42.099 | 5.879 |
| | Russian Federation | | | Brazil | | |
| FDI (% of GDP) | 26 | 1.829 | 1.252 | 43 | 1.973 | 1.503 |
| Portfolio (% of GDP) | 24 | −0.001 | 2.162 | 39 | −0.965 | 1.542 |
| Nominal GDP growth rate | 26 | 1.086 | 6.670 | 43 | 2.856 | 3.463 |
| Gross Fixed Capital Formation (% of GDP) | 26 | 22.533 | 3.854 | 43 | 19.982 | 2.808 |
| Gross Domestic Savings (% of GDP) | 26 | 31.408 | 4.879 | 43 | 20.215 | 3.299 |
| | India | | | | | |
| FDI (% of GDP) | 43 | 0.788 | 0.902 | | | |
| Portfolio (% of GDP | 27 | −1.011 | 1.203 | | | |
| Nominal GDP growth rate | 43 | 6.051 | 2.781 | | | |
| Gross Fixed Capital Formation (% of GDP) | 43 | 28.611 | 6.751 | | | |
| Gross Domestic Savings (% of GDP) | 43 | 23.586 | 7.187 | | | |

Source: Authors calculations from WDI (World Bank).

The average for years from 1963 to 2018 shows that almost all emerging economies had a substantial share of their output as gross domestic savings. However, the average from 1991–2018 shows that Sub-Saharan Africa, South Africa, Russia and Brazil could not hold the domestic savings level and dropped off well below India, China and East Asia in terms of gross domestic savings. These figures are concerning and, to a certain extent, explain the threshold effect of domestic savings on economic

---

[1]　To balance the panel, we did not include other forms of capital inflows such as bank credits, loans and aid, as including these variables would not allow us to conduct panel fixed-effect regression with reasonable number of observations. The panel regression model includes only the BRICS economies and not East Asia and Sub-Saharan Africa.

growth and the level of investments. China and East Asia had are examples of a coupling effect of domestic savings on FDI inflows.

East Asia, Russia, China and Brazil had a higher FDI as a percentage of GDP than India, which as discussed in the introduction, has been changing with gradual reforms in India. More recent summary statistics are presented below, and we can infer from the statistics that following economic reforms, the FDI has become more important as part of the total output of developing economies. Portfolio investment as a percentage of GDP has not changed much in these economies, except for Brazil, where the net portfolio inflow situation has worsened. In terms of GDP growth, only India and Sub-Saharan Africa have registered a 0.7% and 0.5% percentage increase, respectively. All other economies have stagnated.

**Table 2.** Summary statistics (Years: 1991–2018).

| | Sub-Saharan Africa | | | East Asia | | |
|---|---|---|---|---|---|---|
| | Years | Mean | Standard Deviation | Years | Mean | Standard Deviation |
| FDI (% of GDP) | 26 | 2.202 | 0.811 | 26 | 3.262 | 0.785 |
| Portfolio (% of GDP) | 0 | | | 0 | | |
| Nominal GDP growth rate | 26 | 3.841 | 2.059 | 26 | 8.212 | 2.132 |
| Gross Fixed Capital Formation (% of GDP) | 26 | 21.970 | 0.835 | 26 | 38.632 | 3.551 |
| Gross Domestic Savings (% of GDP) | 26 | 21.958 | 2.374 | 26 | 41.295 | 3.933 |
| | South Africa | | | China | | |
| FDI (% of GDP) | 26 | 1.3077 | 1.321 | 26 | 3.663 | 1.170 |
| Portfolio (% of GDP) | 26 | −2.668 | 3.258 | 26 | −0.753 | 0.816 |
| Nominal GDP growth rate | 26 | 2.585 | 1.899 | 26 | 9.763 | 2.309 |
| Gross Fixed Capital Formation (% of GDP) | 26 | 18.719 | 1.943 | 26 | 41.732 | 4.252 |
| Gross Domestic Savings (% of GDP) | 26 | 19.275 | 1.089 | 26 | 44.601 | 4.866 |
| | India | | | Brazil | | |
| FDI (% of GDP) | 26 | 1.283 | 0.849 | 26 | 2.793 | 1.396 |
| Portfolio (% of GDP | 26 | −1.051 | 1.218 | 26 | −1.430 | 1.711 |
| Nominal GDP growth rate | 26 | 6.851 | 1.911 | 26 | 2.497 | 2.736 |
| Gross Fixed Capital Formation (% of GDP) | 26 | 32.507 | 5.669 | 26 | 18.898 | 2.092 |
| Gross Domestic Savings (% of GDP) | 26 | 28.602 | 3.931 | 26 | 18.667 | 2.547 |

Gross fixed capital formation, which is a proxy for domestic investment, decreased by 2.7% in Sub-Saharan Africa, while that in East Asia increased by 1% despite the East Asian Crisis in 1996. It declined in South Africa by 4% and increased in China by 1%. Domestic investment grew by 4% in India and decreased by 1% in Brazil. Gross domestic savings, which are often used as a proxy for domestic availability of capital, decreased by 3% in Sub-Saharan Africa and increased by 2% in East Asia. Domestic savings decreased by 4% in South Africa and increased by 2% in China. It increased by 3% in India and decreased by 2% in India. It is interesting to note that the rise and fall of gross fixed capital and gross domestic savings was similar for both time periods. We can see the threshold income effect of capital inflow: India and China registered a higher growth in the interval between 1991 and 2017 as compared to African countries and Brazil. Per capita income also grew substantially in these periods, and the capital inflows followed.

**Table 3.** Dumitrescu and Hurlin (2012) Granger non-causality test results. Coefficient Z-bar, (.) W-bar.

| Variables | Per Capita GDP (% Annual Change) | FDI (% of GDP) | Portfolio (% of GDP) |
|---|---|---|---|
| Per Capita GDP growth | | | |
| Lags 1 | | 3.7176 *** | 0.9472 |
| | | (2.986) | (1.615) |
| Lags 2 | | 3.4486 *** | 1.348 |
| | | (4.606) | (3.205) |
| Lags 3 | | 2.3933 *** | 2.2200 ** |
| | | (5.215) | (5.413) |
| Lags 4 | | 2.0429 *** | 1.8638 * |
| | | (6.183) | (6.357) |
| FDI (% of GDP) | | | |
| Lags 1 | 3.2539 *** | | 1.0037 |
| | (2.739) | | (1.634) |
| Lags 2 | 1.367 | | 0.6329 |
| | (3.033) | | (2.558) |
| Lags 3 | 0.5578 | | 0.5021 |
| | (3.516) | | (3.550) |
| Lags 4 | 0.0907 | | 0.0885 |
| | (4.096) | | (3.888) |
| Portfolio (% of GDP) | | | |
| Lags 1 | 0.9008 | 7.1996 *** | |
| | (1.569) | (5.553) | |
| Lags 2 | 1.348 | 9.0357 *** | |
| | (3.205) | (10.081) | |
| Lags 3 | 2.2200 ** | 7.9190 *** | |
| | (5.431) | (10.884) | |
| Lags 4 | 1.8638 * | 6.8018 *** | |
| | (6.357) | (12.603) | |
| Obs | 269 | 269 | 150 |

Note: *** $p < 0.01$, ** $p < 0.05$, * $p < 0.1$. Source: Own elaboration from WDI (World Bank).

**Table 4.** Granger bi-directional causality test results: Coefficient is Z-bar and (.) is the W-bar, lags (1).

| | (1) | (2) | (3) | (4) |
|---|---|---|---|---|
| Cause/Effect | GDP growth (% annual) | FDI (% GDP) | Portfolio (% GDP) | Real Exchange Rate (% annual) |
| GDP growth | - | 2.7279 *** (2.4581) | 0.6144 (1.3886) | 0.0138 (1.0098) |
| FDI (% of GDP) | 2.5782 *** (2.3781) | - | 1.0037 (1.6348) | 1.5645 (2.1063) |
| Portfolio (% of GDP) | 0.4882 (1.3088) | 7.1996 *** (5.5534) | - | −0.2977 (0.7895) |
| Real Exchange Rate | 0.3281 (1.2320) | 2.5590 *** (2.8095) | 0.1755 (1.1241) | - |
| Obs | 131 | 131 | 131 | 131 |

Robust standard errors in parentheses: *** $p < 0.01$. Source: Own elaboration from WDI (World Bank).

Per capita GDP growth precedes the movement of the FDI—that is, if per capita GDP growth increases in a year, then the following year, the FDI as a percentage of GDP also increases. This follows the growth effect on investment—a higher foreign capital inflow in fast growing developing economies. A higher income of the average individual of a country incentivizes firms seeking virgin destinations for the demand of their produce. These results support the threshold effect of a certain level of income per capita required to attract foreign capital. Per capita income growth has significant and persistent lag effects in attracting foreign capital with the magnitude of effect decreasing with years.

The FDI also precedes per capita GDP growth, explaining the bi-directional Granger causality. However, the effect is temporary, in that FDI growth does not recurrently cause per capita GDP growth. This can be understood as the capital surges in developing economies during expansionary times. A higher portfolio growth also precedes a higher FDI and has persistent lagged effects. This can be understood from the good capital theory, wherein stable portfolio investment also drives greenfield investments, mergers and acquisitions. This was evident in India 2002–2008 and China 1992–1996, where after the threshold income levels, these countries received stable capital inflows that further augmented FDI flows. Manufacturing and construction sectors in these countries saw huge surges in production, unlike the Latin American and East Asia economies, where short term capital flows led their financial crises in 1994 and 1996, respectively. Firms anticipating short term profits and realizing them in the short run could incentivize investment for a longer term in the form of a tangible foreign direct investment.

The results also point to an interesting causality between portfolio flows and per capita income growth. There is bi-directional causality, with at least three lags between the variables, which signifies that sustained economic growth and stable portfolio flows could cause each other. However, if the portfolio flows are not sustained and stable, then there is no real growth effect. There is, in fact, a negative effect of unstable short-term portfolio flows. Portfolio investment is also significant in determining the real exchange rate movements. Portfolio investment strongly affects the growth of the real exchange rate. This can be understood as the cyclical effect of the portfolio growth of currency appreciation—a higher capital inflow precedes an appreciation in the value of the domestic currency as the demand for the currency increases. There was no such effect with the first lag, implying no immediate effect of portfolio flow on currency appreciation. This could imply that a sustained increase in demand for portfolio stocks of a domestic economy drives down the reserve of that currency in the foreign country, which could then show up in a higher demand for the local currency, thus leading to an appreciation of the domestic economy.

Portfolio investment has a positive effect on the FDI, which is commonly understood as the idea that equity follows debt if the debt if sustained by continued interest and incentives for investment in emerging economies. An appreciation in the real exchange rate leads to more FDI inflows with an intention of reaping surpluses from the value of the currency. With two lags, GDP growth significantly increases the FDI. The FDI is also significant for GDP growth, but the magnitude of the effect is lesser, why is why we do not see the effect on the FDI on per capita income growth after the first lag. Sustained portfolio investment has a larger magnitude effect on the FDI in the second lag, which is fosters the idea that sustained debt flows imply stronger equity flows. Sustained equity and debt flow can also be seen to lead to an increase in the value of currency; this impact is strong, as given by the coefficient of the Z-bar in Table 5. An appreciated currency 'Granger' causes GDP growth at a 10% significance level, which gives leverage to the monetary and fiscal policies of maintaining a stable exchange rate regime for economic growth. A sustained appreciation in currency also leads to higher FDI inflow in search of profitable greenfield investments, mergers, and acquisitions.

**Table 5.** Granger bi-directional causality test results: Coefficient is Z-bar, and (.) is the W-bar, lags (2).

| | (1) | (2) | (3) | (4) |
|---|---|---|---|---|
| Cause/Effect | GDP growth (% annual) | FDI (% GDP) | Portfolio (% GDP) | Real Exchange Rate (% annual) |
| GDP growth | - | 2.5398 *** (3.9199) | −0.5992 (1.464) | 0.313 (2.313) |
| FDI (% of GDP) | 0.8704 *** (2.6579) | - | 0.6239 (2.558) | 1.2634 (3.2634) |
| Portfolio (% of GDP) | 1.3468 (3.204) | 9.0375 *** (10.0818) | - | 6.7604 *** (8.7604) |
| Real Exchange Rate | 1.660 * (3.660) | 1.6369 * (3.6369) | −0.2684 (1.7316) | - |
| Obs | 131 | 131 | 131 | 131 |

Robust standard errors in parentheses: *** $p < 0.01$, * $p < 0.1$. Source: Own elaboration from WDI (World Bank).

## 5. Conclusions

Economic growth is central to attracting capital flow. However, sense of clarity, certainty and predictability in the host nation are musts for investors to make significant sums of investments. This depends on how a host nation interprets its laws for the regulation of capital and whether it respects the property and contractual rights inherent to the investments. An absence of investor confidence, integrity and stable investment regimes exaggerates the perils of investment in host nations and results in lower investments than would otherwise be provided. The state's rule of law and the judiciary's stability of laws and scientific temperament are predictors of clarity, certainty and predictability for investors (Kaufmann et al. 2006). It is also possible for a state to disregard certain social laws and still be the beneficiary of inflows, provided the state provides clarity and transparency and it observes the rule for law where foreign investments are concerned (Sweeney 2010). This is strikingly true in case of India, where the corruption perception index (79/179) and social progress index (93/180) are weak (Rajan and Gopalan 2012). Investors have more confidence in a single party regime, as the commitment of the state is of tremendous importance to investor confidence. The Chinese promulgation of the Equity Joint Venture (EJV) law following the "open door" policy and the opening up different types of economic cooperation in terms of Mergers and Acquisitions (M&A), joint ventures and direct acquisitions are some useful lessons for emerging economies to learn as they go forward to attract foreign capital. The move to continually label sectors as 'permitted' or 'encouraged' may signal to investors a continued desire to attract foreign capital

Increasing dependence on capital flow to sustain economic growth could be detrimental to the macro financial stability of the country, as evidenced by the crises of capital flow in developing economies throughout the world in the past half a century (Taylor 1998). The threshold income effect in attracting foreign capital is an important concern while opening up capital accounts and relying on foreign investment with decreasing emphasis on domestic savings. The success of a capital inflow regime requires looking beyond the regulations at a superficial level, as well as a need to examine the significant differences in how each regime functions and in how each government and agency interacts with that investment regime. Reforms required in both nations go beyond merely looking at the simplification of the investment regimes. The role of the government is crucial in structuring and regulating the national intention with regards to foreign investments and economic growth. For sustained growth in the face of increasing globalization, both China and India would require an increased openness and a continued dissolution of restrictive investment regimes.

There are several steps that each nation could take to improve the situation, including some reforms to restructure investment regimes and some to directly affect the flow of capital. India could benefit by emulating the vertical approval mechanism. It could be useful to be brisk and clear with

respect to guiding investors about relevant decision making and authorities that should be approached for legitimate hassle free investing. Sustained economic growth and stability require a balanced investment regime which caters to required foreign investments comprised of tangible FDI investments and intangible portfolio investments.

There should be no dearth for efficient capital, which, in turn, requires productive investment outlays. To produce the desired results, capital inflows must be channelized to diverse sectors in accordance with needs. Emerging economies could benefit from inflow that provides technology, employment and expertise. The channelization of the inflow should follow productive investments that keep the social, economic, cultural and political fabric of the nation in mind. Investment regimes should not only be concerned about the inflow of capital—they should carefully evaluate the effectiveness and efficiency of the flow given geographic, economic, political and social differences.

**Funding:** This research received no external funding.

**Acknowledgments:** I would like to thank Ramaa Vasudevan, Colorado State University for her insightful comments and reviews on the paper. Thank you Ishita Pradhan, Manuel Cruz, Uthman Baqais, Nina Poerbonegoro and Levi Altringer for your feedback.

**Conflicts of Interest:** The authors declare no conflict of interest.

## Appendix A

**Table A1.** Panel fixed-effect regression.

| VARIABLES | (1) Per Capita GDP Growth | (2) Per Capita GDP Growth | (3) Per Capita GDP Growth |
|---|---|---|---|
| FDI (% of GDP) | 0.420 * | 0.305 * | 0.513 * |
| | (0.192) | (0.110) | (0.229) |
| Gross Capital Formation (% GDP) | 0.0353 | 0.354 ** | |
| | (0.115) | (0.0948) | |
| Gross Domestic Savings (% GDP) | 0.203 | | 0.218 |
| | (0.120) | | (0.179) |
| Net Foreign Assets | −0.003 * | −0.002 * | |
| | (0.0001) | (0.0002) | |
| Portfolio Investment (% GDP) | | −0.188 * | −0.228 ** |
| | | (0.0817) | (0.0787) |
| Constant | −3.052 | −5.472 * | −3.752 |
| | (2.590) | (2.140) | (5.304) |
| Observations | 185 | 150 | 159 |
| R-squared | 0.186 | 0.224 | 0.137 |
| Number of tag | 5 | 5 | 5 |
| VARIABLES | (1) FDI (% GDP) | (2) FDI (% GDP) | (3) FDI (% GDP) |
| Per Capita GDP (% annual growth) | 0.0916 * | 0.0934 * | 0.118 ** |
| | (0.0419) | (0.0383) | (0.0326) |
| Gross Capital Formation (% GDP) | 0.0473 | | |
| | (0.0543) | | |

**Table A1.** *Cont.*

| VARIABLES | (1) Per Capita GDP Growth | (2) Per Capita GDP Growth | (3) Per Capita GDP Growth |
|---|---|---|---|
| Gross Domestic Savings (% GDP) | −0.0421 (0.0865) | −0.00824 (0.0619) | 0.0108 (0.0441) |
| Net Foreign Assets (Current US$ LCU) | 0.003 (0.0011) | 0.002 (0.0023) | |
| Constant | 1.049 (1.455) | 1.365 (1.552) | 1.043 (1.264) |
| Observations | 185 | 185 | 269 |
| R-squared | 0.069 | 0.062 | 0.096 |
| Number of tag | 5 | 5 | 7 |

Robust standard errors in parentheses: ** $p < 0.05$, * $p < 0.1$. Source: Own elaboration from WDI (World Bank).

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
