# Peer review of "Why So Serious about Foreign Capital?"

_ijfs, doi:10.3390/ijfs7030047_

Round 1
Reviewer 1 Report
This is a good and interesting paper.
I have the following general remarks:
- Why are bank credits not analysed, only foreign direct investment and portfolio investment. An explanation might be good.
- Most of the figures in the text are not discussed. In the text there is no reference to the figures.
- Figures 5 and 6: Make the difference clear between M&A and foreign direct investment
Small points:
119: Figure 3: The lines should be similar shown as in the three figures before.
130: Check the number of SOEs.
133: “Foreign Invested Companies Limited by Shares” are not explained, the others are explained.
189: Point is missing.
231: “Schedule I and II” are not explained.
240: “Acquisition” is not needed.
240: Figures 5 and 6 cannot be understood in black and grey as it is now.
390 -394: The text seems to me not so clear. Which year(s)?
397: That investment and domestic saving is similar is for me not so a surprise – please explain.
453: Explain the threshold income effect.
Author Response
Dear Reviewer,
I am grateful for your invaluable time and kind comments on my manuscript. I am thankful that you like the manuscript.
I have made all the changes you suggested especially the literature review and the choice of variables and have also overhauled the paper to make it more sharp and structured.
I hope that you would find the changes appropriate.
I am willing to make more changes as you deem fit
Thank you once again
Sincere Regards

Reviewer 2 Report
Dear Authors,
I thoroughly read and reviewed the manuscript of the article. The methodology and the regression analysis are the strong elements of the paper. The weaknesses of the paper are in the lack of coherence of the selection of the data range and the purpose outlined in the abstract. In addition, there should be explanation of the reasons behind the selection of the key capital inflow variables that affect the economic growth. The limitation of the Granger Casualty Model should also be specified to add clarity to the interpretation of the results.
Comments for Revisions
1. Comments on the Data
· Under the abstract, the author stated that “The study examines the nature and trends of capital inflows in India and China since their economic reforms, especially after the sub-prime crisis 2008.” However, the presented results of the model are for the long data ranges: 1963-2018 and 1991-2018. The author needs to provide clear specification of the data since it critically affects the regression results. Based on the data in the results it is very hard to conclude about the situation post 2008. Moreover, the model results addressed more countries than India and China.
· Perhaps it may help to add a data section with the exact definition, data range (years) specification, and provenance of all data sources.
2. Comments on the key variables in the model
· There are different forms of capital inflows. The paper selects, “FDI inflows, portfolio inflows, real effective exchange rates and GDP growth” as the key variables of study. The author needs to specify the reasons behind selection of FDI and Portfolio inflows. The issue is particularly relevant in making conclusions about capital inflows in Sub-Saharan African Countries since there are studies that indicate that the two forms of inflows are not dominant forms of inflows in those countries. Moreover, FDI and portfolio inflows are new to the Sub-Saharan African since the conventional inflows take the forms of loans and aid.
3. Limitations of the Granger Casualty Model
· Although the article has robustness check, it should also explain the limitation of the model. Granger Casualty model has several limitations. Therefore, the author needs to explain the limitations of the model to add clarity to the interpretation of the results.
· There are different variables that affect GDP growth rates. The robustness check model only includes factors that are related to domestic capital: capital formations and savings.
Author Response

(The authors gave the same response as above.)
